# The CM SAF R Toolbox—A Tool for the Easy Usage of Satellite-Based Climate Data in NetCDF Format

**Steffen Kothe** [1,*] , **Rainer Hollmann** [1] , **Uwe Pfeifroth** [1] , **Christine Träger-Chatterjee** [2] **and Jörg Trentmann** [1]

1   Deutscher Wetterdienst, 63067 Offenbach, Germany; Rainer.Hollmann@dwd.de (R.H.); Uwe.Pfeifroth@dwd.de (U.P.); Joerg.Trentmann@dwd.de (J.T.)
2   EUMETSAT, 64295 Darmstadt, Germany; Christine.Traeger@eumetsat.int
*   Correspondence: Steffen.Kothe@dwd.de

**Abstract:** The EUMETSAT Satellite Application Facility on Climate Monitoring (CM SAF) provides satellite-based climate data records of essential climate variables of the energy budget and water cycle. The data records are generally distributed in NetCDF format. To simplify the preparation, analysis, and visualization of the data, CM SAF provides the so-called CM SAF R Toolbox. This is a collection of R-based tools, which are optimized for spatial data with longitude, latitude, and time dimension. For analysis and manipulation of spatial NetCDF-formatted data, the functionality of the cmsaf R-package is implemented. This R-package provides more than 60 operators. The visualization of the data, its properties, and corresponding statistics can be done with an interactive plotting tool with a graphical user interface, which is part of the CM SAF R Toolbox. The handling, functionality, and visual appearance are demonstrated here based on the analysis of sunshine duration in Europe for the year 2018. Sunshine duration in Scandinavia and Central Europe was extraordinary in 2018 compared to the long-term average.

**Keywords:** NetCDF; satellite-based; climate monitoring; sunshine duration; R-project

## 1. Introduction

Satellite-based data have been used for several decades to continuously observe the Earth's atmosphere and surface. In addition, the quality of satellite-based data steadily increases due to improved retrieval algorithms, more sophisticated computational models, and better satellite instruments. Nowadays, there are satellite-based climate data records available for many essential climate variables [1], which cover time periods of more than 35 years. For these reasons satellite-based data records gain more and more importance for climate monitoring purposes.

The EUMETSAT Satellite Application Facility on Climate Monitoring (CM SAF) develops, generates, archives, and distributes high-quality satellite-derived data records of the energy budget and water cycle to monitor, understand, and adapt to climate variability and climate change. The product portfolio of the CM SAF includes regional and global variables for surface radiation, sunshine duration, cloud fractional coverage, cloud properties, water vapor, evaporation, precipitation, surface albedo, and land surface temperature. The data records are based on sensors onboard geostationary and polar-orbiting satellites. The spatial resolutions of CM SAF climate data records range from 0.5 degrees to higher than 0.05 degrees and the temporal resolution ranges from monthly up to 15 min. These high resolutions in space and time, in combination with long time series (earliest data start in 1982), result in huge data amounts.

Common data formats for geospatial data records are NetCDF (network common data format), HDF (hierarchical data format) or GRIB (gridded binary). These data formats have the advantage to

store large amounts of geospatial data in an efficient manner. In addition, there is the possibility to include meta-data, such as the sensing instrument, the version of the algorithm applied, or uncertainty measures, which is essential for climate data records. In the climate community NetCDF acts as quasi standard (e.g., most data from the NOAA Climate Data Record Program [2] are provided in NetCDF, climate model output is usually in NetCDF format [3], ESA CCI data are provided in NetCDF format [4]) and CM SAF climate data records are usually provided in NetCDF format.

NetCDF (network common data format) is a set of interfaces for array-oriented data access including a collection of data access libraries for languages such as C, Fortran, C++, Java, or R. NetCDF is a machine-independent format for representing scientific data [5]. NetCDF files are self-describing because the files include meta-data. The file format is portable and can be used on many different computer systems. NetCDF data are easily scalable as subsets of a dataset can be accessed efficiently [5]. In addition, NetCDF files offer several options for compression and can be archived very efficiently.

Due to huge data amounts, the access management and handling of satellite-based climate data records can be challenging. For the easy processing of NetCDF formatted files it is essential to have efficient tools to generate spatial and temporal subsets of climate data records and to process the data. The analysis of climate data requires tools that can handle the spatial and temporal components of the data. In addition, the visualization of NetCDF data is important in climate analysis and for presenting results. These tools have to be as efficient as possible and easily accessible for a wide range of users across platforms.

Several tools are already available to facilitate the usage of NetCDF data. Command line-based tools, such as the climate data operators (CDO) [6] or the NetCDF operators (NCO) [7] are commonly used in the scientific community. CDO offers a collection of about 600 command line operators, which are designed for manipulation and analysis of NetCDF-formatted climate data and data from numerical weather prediction models. CDO includes operators, which are specifically implemented for spatial and temporal analysis of climate data, such as means, variabilities, standard deviations, or percentiles. Compared to CDO, NCO is a more general tool for manipulating NetCDF files, including their structure and meta-data. NCO is not specifically designed for climate analysis. Besides command-line-based solutions there are software tools with a graphical user interface (GUI), such as GIS-based tools (geographic information system). GIS combines the analysis and visualization of data within GUI-based software. There are several free and commercial GIS tools, which can handle NetCDF data, although not specifically designed for this data format. Another group of software tools, such as Panoply [8] or ncview [9] were purely developed to visualize NetCDF data and its meta-data. These tools offer several options to display NetCDF data, but no or only very basic operators for data analysis.

As all of the above-mentioned tools have their advantages and limitations, it is very common in the climate science community to develop individual tools, which are based on scientific computing languages, such as IDL, Python, or R. These languages provide extensions or libraries which allow importing of NetCDF data. However, analyzing and visualizing climate data records in NetCDF format in this way requires at least basic programming skills. Furthermore, existing tools and scripts are usually not easily accessible for a wider number of potential users.

Satellite-based climate data are used in a wide range of applications, such as climate monitoring, climate model evaluation, for the estimation of the solar energy potential, or for applications in the field of agriculture, tourism, or health. Therefore, many people from different fields of interest and with different educational backgrounds are interested in satellite-based climate data. Thus, it is important to provide tools which allow easy access to satellite-based climate data in NetCDF format. In this paper we present an R-based toolkit for preparation, analysis, and visualization of satellite-based climate data records—the CM SAF R Toolbox. The intention, operators and functionalities of the CM SAF R Toolbox are described in Section 2. Section 3 shows the application of the toolbox based on an example for CM SAF sunshine duration data. Section 4 gives conclusions and an outlook of possible future developments and applications.

## 2. The CM SAF R Toolbox

Data of in situ observations are generally stored in the form of ascii files instead of NetCDF, and many numerical weather prediction models deliver output in grib format. However, this is not feasible for satellite-based climate data, as they cover large geographical areas and; therefore, have much larger file sizes than in situ data. The handling of satellite-based climate data differs from the handling of in situ data. As a consequence, there is a huge demand for satellite-based climate data training for National Meteorological and Hydrological Services, Regional Climate Centers, universities, and research institutes. The European Organisation for the Exploitation of Meteorological Satellites (EUMETSAT) provides regular climate training workshops, which are supported by CM SAF. The main goal of these training workshops is to instruct and support (potential) users in applying satellite-based climate data. The CM SAF R Toolbox (further referred to as the toolbox) was originally developed in 2015 to support these training activities with an easy-to-use tool, which includes all necessary steps to get started with satellite-based data in NetCDF format.

Additional requirements for the toolbox are a platform independent and easy installation, adaptability to individual user needs, and the possibility to expand it to individual tasks. The first choice to fulfill all these needs was the scientific programming language R [10], which is a programming language for statistical computation and visualization. The R system includes a language and a run-time environment for graphics, debugging, access to certain system functions, and the ability to run programs, which are stored in script files [10]. In addition, it is easily expandable by a variety of additional packages from a huge community, which cover a large range of applications [11]. Besides functioning as a climate training tool, the main intention of the CM SAF R Toolbox is to be software which allows every user an easy application of CM SAF NetCDF climate data with free and open software.

The toolbox can be downloaded for free from the CM SAF webpage [12]. It is provided as a zip file including a set of R-scripts, example data for testing and documentation. As the toolbox is based on R, it requires the installation of the R system, which is available for all common platforms and operating systems [13]. In addition, it is recommended to install the free RStudio [14] software, which increases the clarity of R code and supports R-shiny (see Section 2.3) applications. However, RStudio is not necessarily required to run the CM SAF R Toolbox.

The development of the CM SAF R Toolbox started with a set of R-scripts for data analysis and visualization in combination with the functionality of the cmsaf R-package. Further development has included the increase and extension of operators of the cmsaf R-package, improved and easier usage and the integration of all necessary steps to work with CM SAF data. These steps include three main tasks:

1. Data preparation
2. Analysis of climate data
3. Visualization of data and results.

These tasks are described in more detail in Sections 2.1–2.3. To facilitate these tasks, the toolbox comes with a pre-defined file structure, which is illustrated in Figure 1.

### 2.1. Data Preparation

CM SAF data can be ordered free of charge via a web user interface (https://wui.cmsaf.eu). Ordered data are delivered via sftp (secure file transfer protocol) or via https in the form of tar files. A tar file (tape archiver) or tarball is a file format to archive data on Linux systems (but can also be read on Windows and Mac systems). Each tar file contains a set of NetCDF data files (one NetCDF file per time step). The CM SAF R Toolbox supports users to progress from the downloaded tar file to a ready-to-use NetCDF file. This step includes the extraction of all tar files of each order, an optional restriction of the time range and spatial extension, and the merging of all time steps into one NetCDF file. The latter is important to increase the ease of time series analysis.

The data preparation is done by the R-script *Prep.Data.R*. This is a ready-to-use script, which can be started in R or RStudio without any changes of the code. The user is guided through the preparation process and can give optional inputs or use default values. The result of this process is one NetCDF file that includes all time steps for the chosen time period and region. This file will be written into the output folder that comes with the CM SAF R Toolbox (see Figure 1).

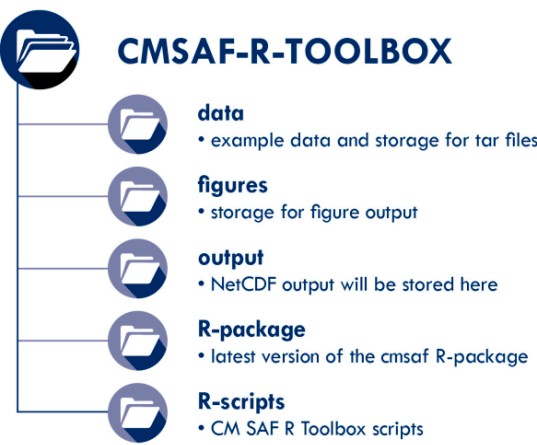

**Figure 1.** CM SAF R Toolbox folder structure.

*2.2. Data Analysis*

The most important feature of the CM SAF R Toolbox is the script *Apply.Function.R*, which is an interface to the functionality of the cmsaf R-package [15]. This script can be started in R or RStudio without any code changes and guides the user through the process. Usually this step is the second step after preparing the downloaded tar file as described in Section 2.1. The user can choose a NetCDF file by help of a file browser. The main task of the *Apply.Function.R* script is to get all necessary information to apply an operator of the cmsaf R-package. This information includes the variable name, the name and path of the input and output file and the name of the operator. The user can choose from a list of more than 45 operators. After applying an operator, the result is written into a NetCDF file which will be placed into the output folder of the CM SAF R Toolbox.

The cmsaf R-Package

R-packages are fundamental units of reproducible R code, which include reusable R functions, the documentation that describes how to use them, and sample data [16]. The cmsaf R-package is a collection of R operators for the analysis and manipulation of geospatial NetCDF data with a time dimension. The package was developed and tested for CM SAF climate data records, but it is applicable for many other gridded NetCDF data that follow CF conventions [17]. So far, the cmsaf R-package consists of more than 60 operators (Tables 1 and 2). There is a common syntax for the application of most of the cmsaf operators. The user has to give the name of the operator, the name of the variable, an input file, and an output file name. All operators and their usage are described in detail in the documentation of the cmsaf-package [15]. Most of the cmsaf operators can be easily applied by help of the script *Apply.Function.R.* The user simply chooses an operator and the script provides all necessary inputs. An overview of all operators currently available in the toolbox is provided in the tables below (Tables 1 and 2).

**Table 1.** Operators for data analysis of the cmsaf R-package (version 1.9.5). Operators that are included in the CM SAF R Toolbox are marked in italics.

| Operator | Description | Operator | Description |
|---|---|---|---|
| cmsaf.add | Add fields of two files. | *seasmean* | Seasonal means. |
| *cmsaf.addc* | Add constant to data. | *seassum* | Seasonal sums. |
| cmsaf.div | Divide fields of two files. | *timmax* | All-time maxima. |
| *cmsaf.divc* | Divide data by constant. | *timmean* | Mean of time series. |
| cmsaf.mul | Multiply fields of two files. | *timmin* | All-time minima. |
| *cmsaf.mulc* | Multiply data with constant. | *timpctl* | Percentile over all time steps. |
| cmsaf.sub | Subtract fields of two files. | *timsd* | All-time standard deviations. |
| *cmsaf.subc* | Subtract constant from data. | *timsum* | Sum of time series. |
| *dayrange* | Diurnal range. | *trend* | Linear trends. |
| *divdpm* | Divide by days per month. | *wfldmean* | Weighted spatial mean. |
| *fldmax* | Field maximum. | *ydaymean* | Multi-year daily means. |
| *fldmean* | Field mean. | *year.anomaly* | Annual anomalies. |
| *fldmin* | Field minimum. | *yearmean* | Annual means. |
| *mon.anomaly* | Monthly anomalies. | *yearsum* | Annual sums. |
| *monmax* | Monthly maxima. | *ymonmax* | Multi-year monthly maxima. |
| *monmean* | Monthly means. | *ymonmean* | Multi-year monthly means. |
| *monmin* | Monthly minima. | *ymonmin* | Multi-year monthly minima. |
| *monsd* | Monthly standard deviation. | *ymonsd* | Multi-year monthly standard deviations. |
| *monsum* | Monthly sums. | *ymonsum* | Multi-year monthly sums. |
| *muldpm* | Multiply by days per month. | *yseasmax* | Multi-year seasonal maxima. |
| *multimonmean* | Multi-monthly means. | *yseasmean* | Multi-year seasonal means. |
| *multimonsum* | Multi-monthly sums. | *yseasmin* | Multi-year seasonal minima. |
| *seas.anomaly* | Seasonal anomalies. | *yseassd* | Multi-year seasonal standard deviations. |

**Table 2.** Operators for data manipulation of the cmsaf R-package (version 1.9.5). Operators that are included in the CM SAF R Toolbox are marked in italics.

| Operator | Description | Operator | Description |
|---|---|---|---|
| *box_mergetime* | Combine files and simultaneously cut a region. | *remapbil* | Bilinear grid interpolation. |
| change_att | Change attributes of variable. | *sellonlatbox* | Select region by longitude/latitude. |
| extract.level | Extract levels from four-dimensional variables. | *selmon* | Extract list of months. |
| *extract.period* | Remove time period. | *selperiod* | Extract list of dates. |
| get_time | Convert time steps to POSIXct. | *selpoint* | Extract data at given point. |
| *levbox_mergetime* | Combine files and simultaneously cut a region and level. | selpoint.multi | Extract data at multiple points. |
| *ncinfo* | Get information about file content. | seltime | Extract specific time step. |
| read_ncvar | Read variable. | *selyear* | Extract list of years. |

## 2.3. Data Visualization

The CM SAF R Toolbox can be used to visualize NetCDF data. This is done by the R script *CMSAF_Visualizer.R* which uses a R-shiny based application. R-shiny is an R package that makes it easy to build interactive web applications without knowledge of HTML or JavaScript [18]. The *CMSAF_Visualizer.R* can be used to create interactive plots of 2D spatial maps or 1D time series. There are multiple options to adapt the plot to the user's need (Figure 2). This includes the adaption of, for example, color bars, the scale range, the scale color number, the line style, the color for 1D time series plots, or the text for axis captions and the title. In addition, there are options to change the projection, to add the name of specific locations, or to plot station and satellite data simultaneously. Besides the ability of plotting the data, the *CMSAF_Visualizer.R* provides information about the data in the form of statistics, histograms, and lists of meta-data. The output can be saved as a png file (portable network graphics).

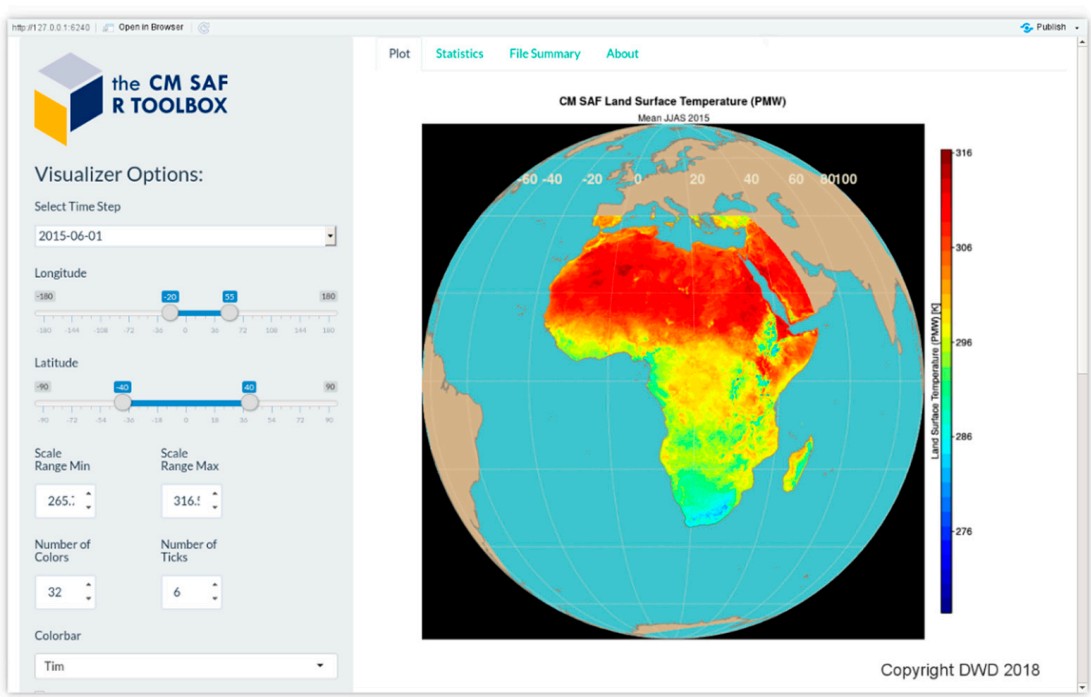

**Figure 2.** The interface of the R-shiny based *CMSAF_Visualizer.R* application.

## 2.4. CM SAF R Toolbox Interface

For convenience, the data preparation, analysis, and visualization can be controlled by one single R-shiny based application, which gives the look and feel of a graphical user interface (GUI). After starting the script *CMSAF-R-TOOLBOX.R*, a window pops up where the user can choose between the three main functionalities (Figure 3). This application provides a brief description of the toolbox functionalities and a brief manual for each of the three steps.

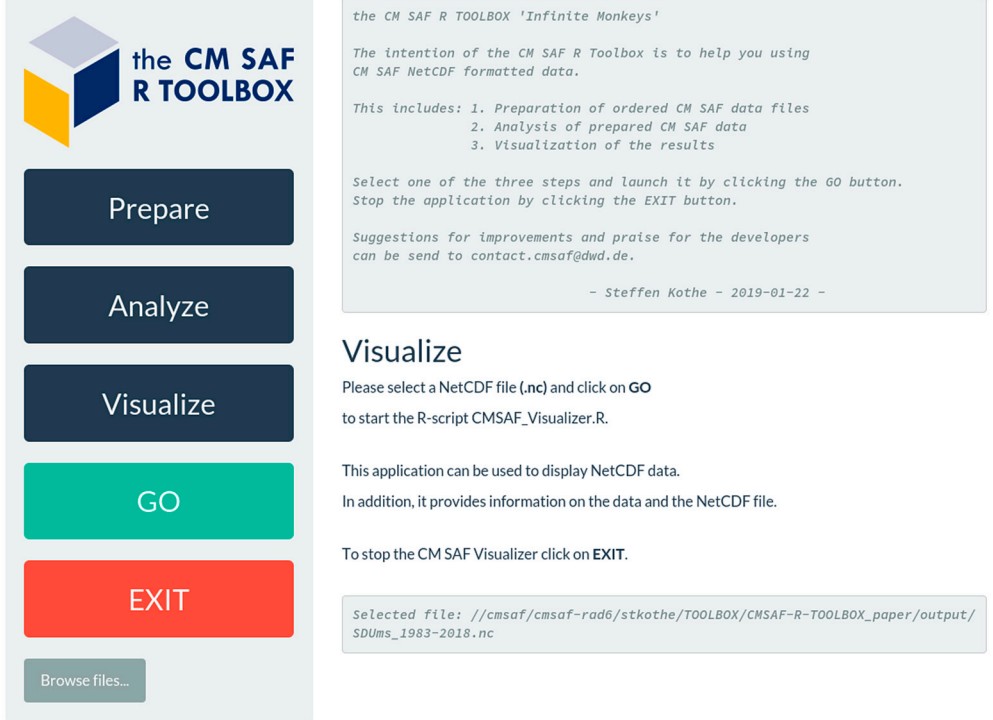

**Figure 3.** Main user interface of the CM SAF R Toolbox.

## 3. Example Application for Sunshine Duration

In this section the capabilities of the CM SAF R Toolbox will be demonstrated for the analysis of sunshine duration for Europe in 2018. The year 2018 was a year with extraordinary occurrences of extreme weather events, such as drought and heat, in many parts of the world [19]. To investigate the situation for sunshine duration in Europe, the satellite-based sunshine duration for 2018 and the corresponding climate data record are analyzed and discussed below.

### 3.1. CM SAF Sunshine Duration Data

The CM SAF provides the SARAH-2.1 (Surface Solar Radiation Data Set–Heliosat, version 2.1) climate data record for solar surface radiation. SARAH-2.1 includes the parameters: Effective cloud albedo (CAL), surface incoming shortwave radiation (SIS), surface direct irradiance (SDI), spectrally resolved irradiance (SRI), and sunshine duration (SDU) [20]. The SDI includes two direct radiation components, the surface incoming direct radiation (SID) and the direct normalized irradiance (DNI). The basis for the retrieval of satellite-based SDU are the SARAH-2.1 30 min instantaneous DNI data and the World Meteorological Organization (WMO) threshold for bright sunshine, which is defined by $DNI \geq 120 \text{ W/m}^2$. Daily SDU is derived using the ratio of instantaneous SARAH-2.1 DNI (slots) exceeding the DNI threshold, and hence are considered as sunny slots, to all slots during daylight. In addition, the sunshine duration for each pixel is weighted depending on the number of surrounding cloudy and sunny grid points for two successive time steps [21]. Details on the retrieval and evaluation of SARAH-2.1 SDU can be found in Kothe et al. [21], the SARAH-2.1 Algorithm Theoretical Baseline Document and the SARAH-2.1 validation report [20].

The SARAH-2.1 SDU Thematic Climate Data Record (TCDR) [20] is available as daily or monthly sums for the time period 1983 to 2017. In addition, the CM SAF provides an Interim Climate Data Record (ICDR) [22] based on the SARAH-2.1 algorithm [23]. For the CM SAF, an ICDR denotes a regularly updated TCDR in shorter time latency, with an algorithm and processing system as consistent as possible to the generation of the reference TCDR. This definition follows the definition of a TCDR as outlined in Dowell et al. [24]. Here, this ICDR is provided with a delay of a few days after the satellite measurement and starts in January 2018.

### 3.2. SDU Data Preparation

This study applied daily and monthly sums of the SDU TCDR for the time period 1983 to 2017 and the SDU ICDR for 2018. Both cover the full Meteosat prime disk (Figure 4a). In total this would be about 225 GB (daily) or 7.6 GB (monthly) in NetCDF4 format. To reduce the amount of data, the spatial domain was restricted to Europe during the ordering process (see Figure 4b). For the chosen region, the final amount of SDU data was about 30 GB (daily) or 800 MB (monthly). After ordering the data (https://wui.cmsaf.eu), the user gets a confirmation mail and finally a mail including download links via sftp or https. The data are provided as tar files with a maximum size of 4 GB. As the data amount of this order exceeded this maximum size, the order was divided into several tar files.

After downloading the data, the first step was to prepare the data for the analysis. This step required: (a) source the R-script *CMSAF-R-TOOLBOX.R*, (b) to choose one of the tar files, (c) to start the prepare step, and (d) to follow the given instructions. The script extracts the information for the included time range, spatial dimensions, and names of variables from all tar files with matching order number, which optionally can be chosen by the user. The preparation of the data includes the extraction of the tar files, the optional restriction of time and space, and the merging of all time steps into one NetCDF file. For the SARAH-2.1 SDU TCDR data, the tar files included one NetCDF file for each time step (in total 12,784 daily files), which were merged into one file. For the SDU ICDR the final NetCDF file included 365 time steps for the entire year 2018.

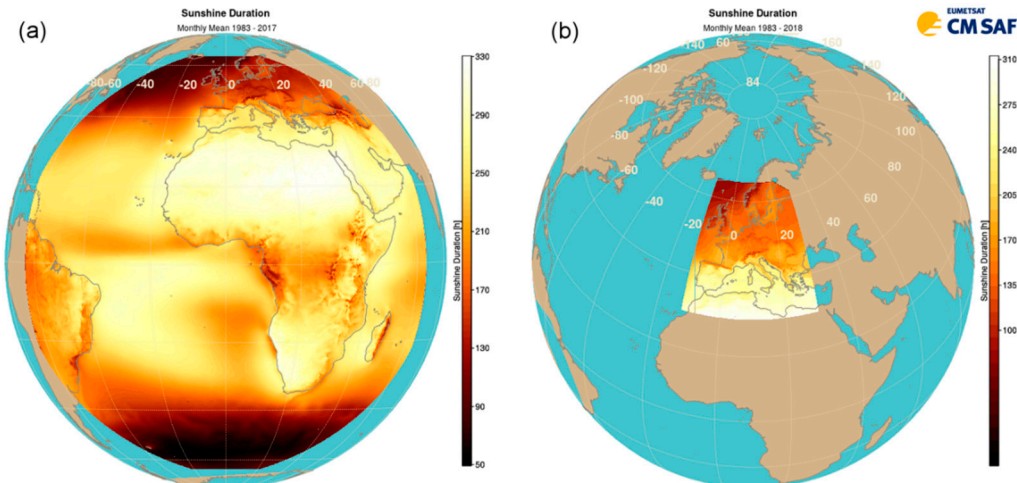

**Figure 4.** (**a**) Mean monthly sunshine duration (1983–2017) for the whole Meteosat disk; and (**b**) mean monthly sunshine duration (1983–2018) for the chosen region of interest in orthographic projection.

### 3.3. Sunshine Duration Analysis for Europe 2018

To get a first impression of SDU in 2018 in Europe, spatial plots of the annual sum for 2018 and the mean annual sum for the time period 1983 to 2017 were created (see Figure 5). The annual sum for 2018 was derived by applying the *timsum* operator to the SDU daily means of 2018. The mean annual sum for the SDU TCDR from 1983 to 2017 was derived by applying (1) the *yearsum* operator to the SDU monthly sums for this period and (2) the *timmean* operator. The results were visualized using the *CMSAF_Visualizer.R*.

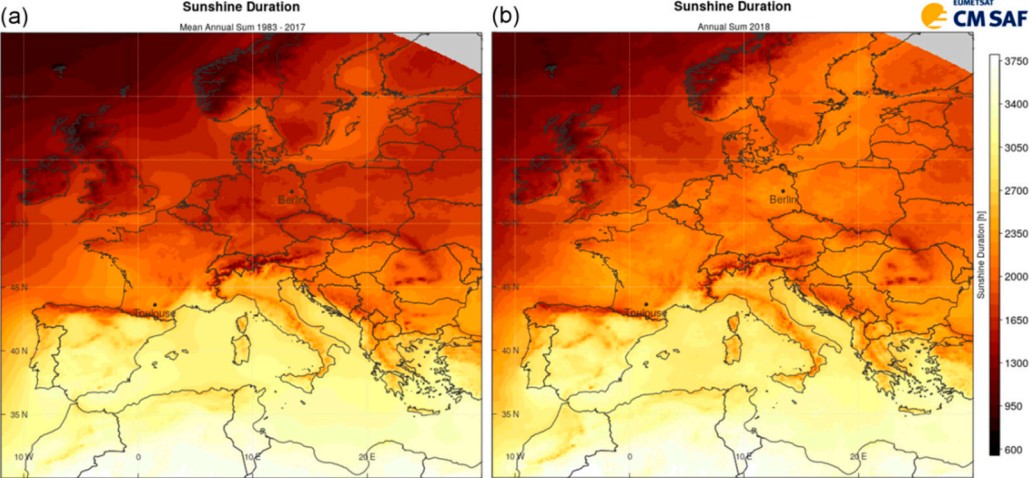

**Figure 5.** Sunshine duration mean annual sum for the time period 1983 to 2017 (**a**) and for the year 2018 (**b**). The cities of Berlin and Toulouse are highlighted.

SDU in Europe is usually highest in the Mediterranean and the Iberian Peninsula, decreasing to the north and especially to the northwest, with lowest values on the British Islands and Scandinavian mountains. This pattern is mainly due to the day length and the cloud coverage. The color bar in both plots of Figure 5 was the same, making it easy to compare them. Figure 5 shows that the annual SDU for 2018 south of 45°N looked very similar to the long-term mean sum (Figure 5a). The most obvious differences were in Central Europe, the Baltic Sea and Scandinavia, where the values were much higher than the mean. These differences could be further investigated by looking at spatial anomalies of SDU.

The CM SAF R Toolbox and the included cmsaf R-package provide operators to create annual, seasonal, and monthly anomalies. The operators *year.anomaly*, *seas.anomaly,* and *mon.anomaly* were

applied to annual, seasonal, and monthly sums of SDU for the time period 1983 to 2018, which were derived by applying the *yearsum*, *seassum*, and *monsum* operators, respectively. The resulting files were visualized by the *CMSAF_Visualizer.R* (see Figure 6). To highlight positive and negative anomaly values, a blue–whitered color bar was chosen and the values were symmetrically centered around zero. Figure 6 shows absolute values of the SDU anomalies, which is why the values decreased from annual to seasonal and monthly. To have the highest contrast, different ranges of the color bar were chosen for these figures. For the mean of May 1983 to 2017, Figure 6d presents the mean monthly SDU sum, which was derived by applying first the *monsum* operator and then the *ymonmean* operator to SDU daily sums from 1983 to 2017.

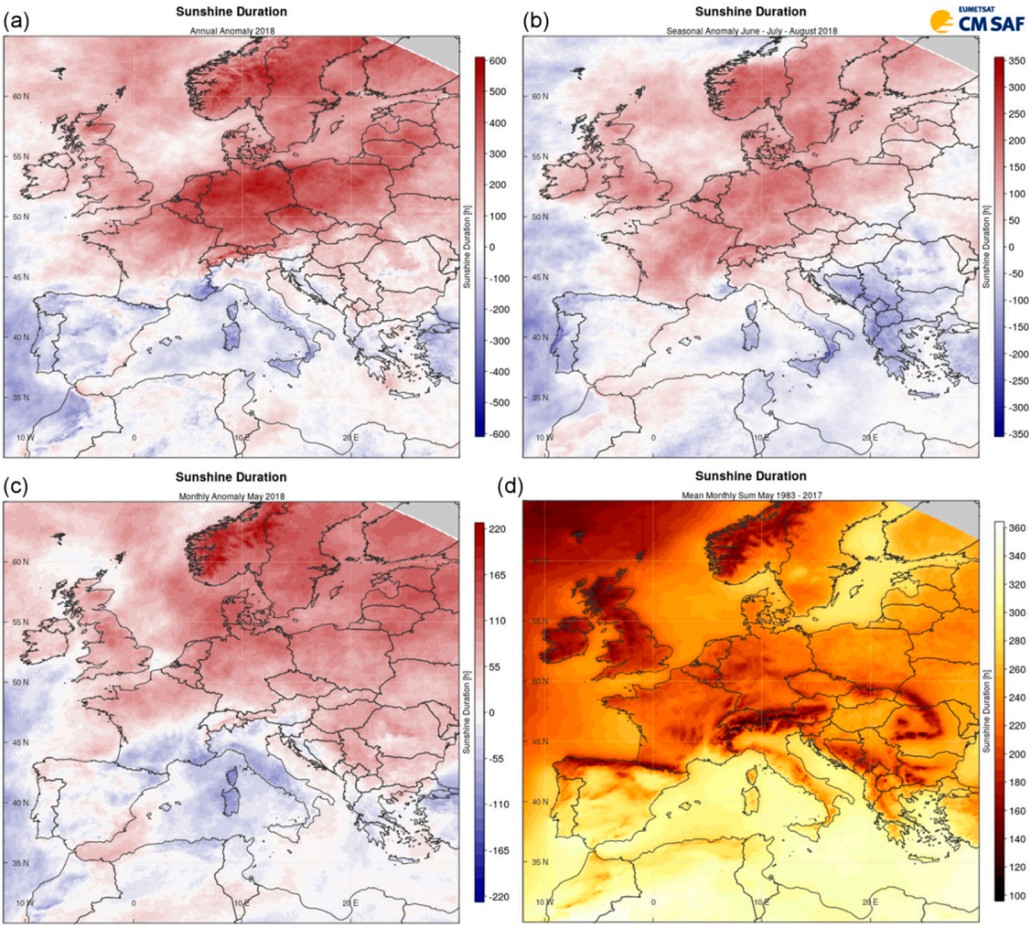

**Figure 6.** Annual sunshine duration (SDU) anomaly for 2018 (**a**), seasonal SDU anomaly for summer 2018 (**b**), monthly SDU anomaly for May 2018 (**c**), and mean monthly SDU sum for May 1983 to 2017 (**d**).

The annual SDU anomaly shows the deviation of the annual mean SDU sum from the long-term annual SDU sum. Figure 6a reveals a strong gradient in the 2018 SDU anomaly from Northern to Southern Europe. There is a strong positive SDU anomaly in regions north of about 45°N and no, or even slightly, negative anomaly values in the Mediterranean and Northern Africa. The highest positive values were in Germany and Scandinavia, with up to 600 h above the long-term mean.

The seasonal SDU anomaly shows the deviation of the seasonal mean SDU sum from the long-term seasonal SDU sum. There was a very strong positive seasonal anomaly in summer 2018 in Central Europe, Scandinavia, and the British Islands, which reached values of up to 200 h above average (Figure 6b). Negative seasonal anomalies could be found in Portugal, Southern Italy, and the Balkans.

The monthly SDU anomaly shows the deviation of the monthly SDU sum from the long-term monthly SDU sum. In Figure 6c it can be seen that in May 2018 SDU was extraordinarily high in

many parts of Europe. The long-term mean monthly SDU sum for the Scandinavian mountains was about 170 h (Figure 6d). The monthly SDU sum in May 2018 in this region was about 230% above the long-term value.

To have a more detailed look into the special features of SDU, the SDU time series for Berlin and Toulouse were compared (Figures 7 and 8). Berlin was situated in a region with high positive SDU anomalies in 2018 (Figure 7). Toulouse had a similar long-term mean annual SDU sum as the mean SDU sum for 2018 in Berlin. Both cities are marked in Figure 5. The basis for the plots in Figure 7 was the monthly sums of SDU for the time period 1983 to 2018. To extract the grid points of Berlin (52.5°N; 13.4°E) and Toulouse (43.6°N; 1.45°E) the *selpoint* operator was used. Figures 7 and 8 are standard plots, which were created with the *analyze timeseries* option in the *CMSAF_Visualize.R*. The time series of monthly sums are shown including a linear trend line (a); the annual cycle including maximum, minimum, and mean (b); the monthly anomalies including the linear trend (c); a box plot of the annual cycle (d); the annual monthly means including the overall mean (e); and the SDU histogram (f).

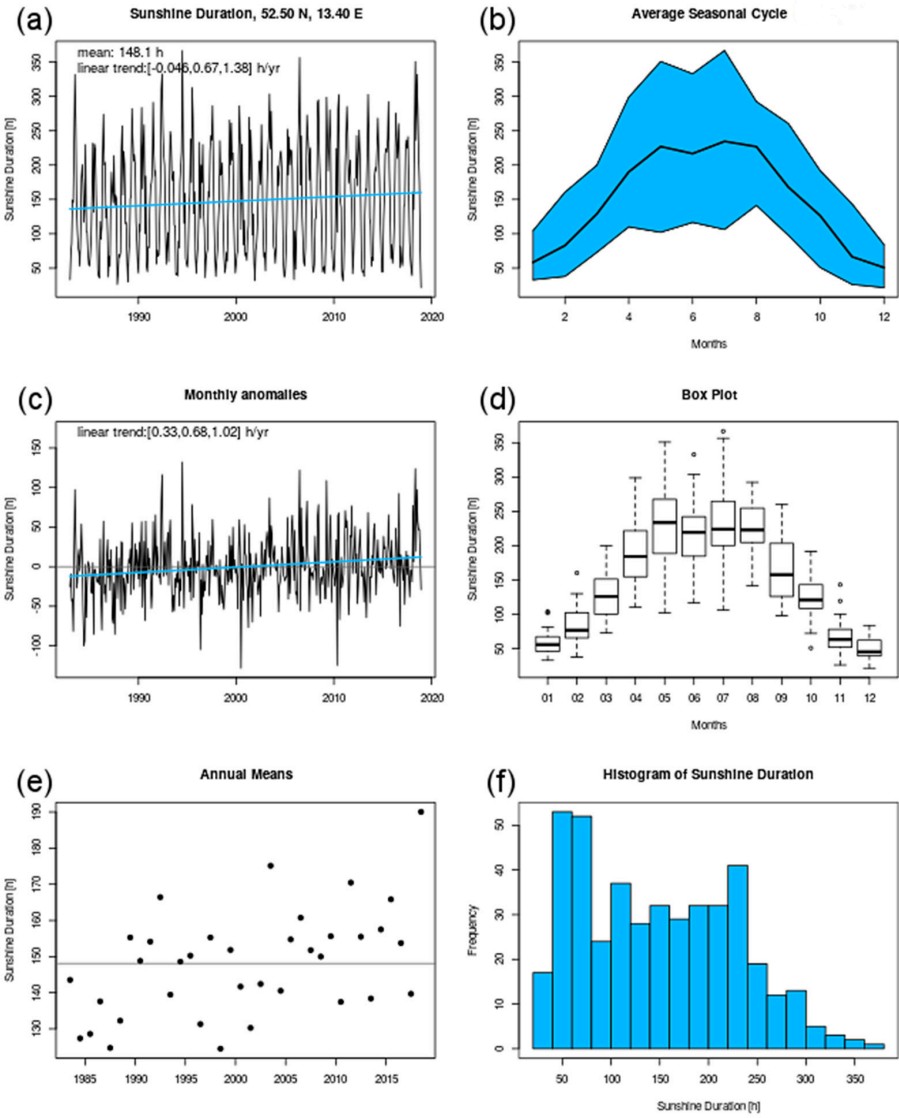

**Figure 7.** Analysis plots for the SDU time series for Berlin (52.5°N; 13.4°E) from 1983 to 2018. Shown are the time series of monthly sums including a linear trend line (**a**); the annual cycle including maximum, minimum, and mean (**b**); the monthly anomalies including the linear trend (**c**); a box plot of the annual cycle (**d**); the annual monthly means including the overall mean (**e**); and the SDU histogram (**f**).

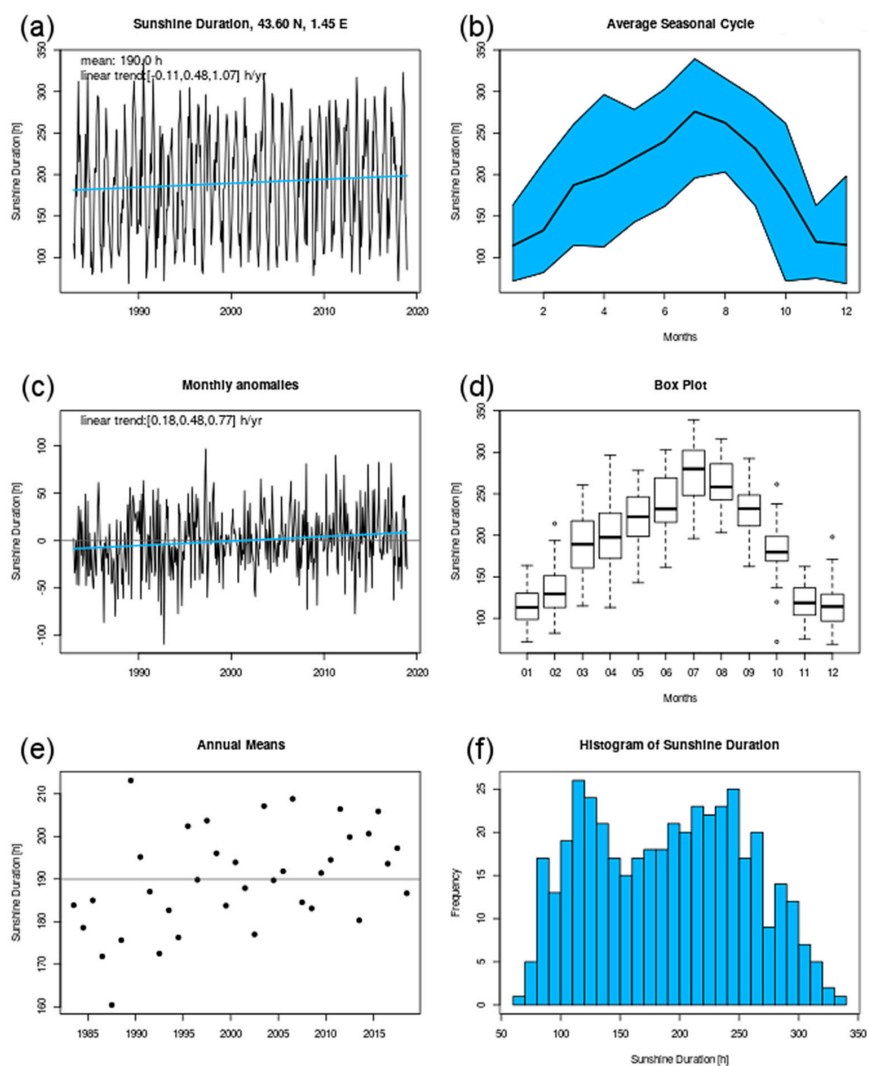

**Figure 8.** Analysis plots for the SDU time series for Toulouse (43.6°N; 1.45°E) from 1983 to 2018. Shown are the time series of monthly sums including a linear trend line (**a**); the annual cycle including maximum, minimum, and mean (**b**); the monthly anomalies including the linear trend (**c**); a box plot of the annual cycle (**d**); the annual monthly means including the overall mean (**e**); and the SDU histogram (**f**).

The 2018 SDU monthly means for Berlin (Figure 7a) were very high, with values up to 350.8 h in May 2018. Although this was the highest May value of the whole time series, there were higher monthly sums in July 2006 (358.4 h) and July 1994 (366.7 h). The annual cycle (Figure 7b) for Berlin was quite symmetrical, with strong variability from April to August, while the annual cycle of Toulouse (Figure 8b) showed less variability and a maximum that was shifted to July and August. The extraordinary high SDU in Berlin in 2018 was also confirmed by the time series of monthly anomalies in Figure 7c, while the SDU monthly anomalies in Toulouse (Figure 8c) in 2018 were close to zero. The annual mean for 2018 in Berlin (Figure 7e) was by far the highest in this time series. However, this record sunshine duration sum was almost the mean annual SDU for Toulouse (Figure 8e).

By application of the *ydaymean* and the *timsum* operator, it could be investigated as to when the accumulated SDU for a year reached the long-term annual mean. For Berlin this was the case at 28 August 2018, where the accumulated daily SDU for 2018 (1775.1 h) exceeded the climatological annual mean sum (1771.7 h). This means that even if in the months of September to December 2018 there would have been no sunshine, the annual sum would still correspond to the long-term mean annual sum.

The results for Berlin were representative of Central Europe, which was affected by a strong positive SDU anomaly in 2018. By application of the *sellonlatbox* operator to the monthly sums of SDU (see Figure 9), a large part of Central Europe was extracted from the data and; subsequently, spatially averaged with the *fldmean* operator. The resulting time series was visualized with the *CMSAF_Visualizer.R* (Figure 10).

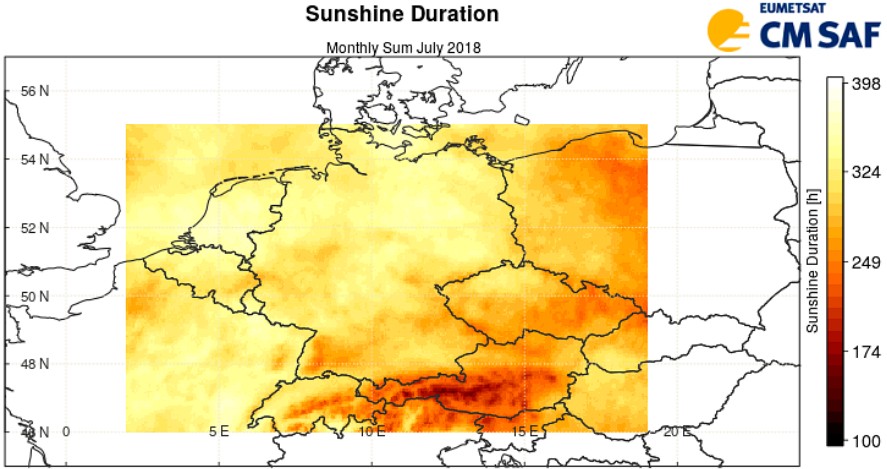

**Figure 9.** A spatial subset of SDU monthly sum for July 2018 for Central Europe.

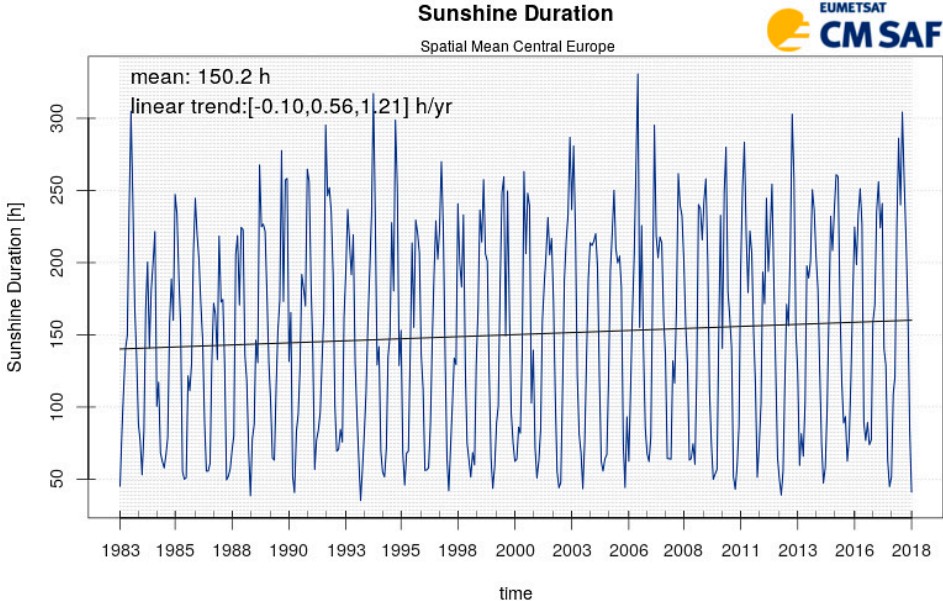

**Figure 10.** Spatial mean of SDU monthly sums from 1983 to 2018 for Central Europe.

Figure 10 shows a similar temporal evolution of SDU as Berlin, with a slightly increasing trend and a similar mean of about 150 h. Although there were several years with higher monthly peaks, the annual mean monthly SDU sum for Central Europe was highest in 2018 (not shown) with 175.2 h. This value was slightly higher than for the year 2003 (174.9 h), during which Europe also experienced an extremely hot, sunny, and dry summer [25].

*3.4. Summary of SDU Analysis*

The analysis of satellite-based CM SAF SDU climate data records for the time period 1983 to 2018, in Section 3.3, revealed that in 2018 SDU was extraordinarily high in Central and Northern Europe. It was shown that, based on the 35 years of CM SAF SARAH-2.1 data, 2018 had the highest recorded

annual SDU in these regions. The month with the highest positive anomaly was May, which was, in many regions, even the month with the highest absolute SDU in 2018. The highest recorded monthly sums in Berlin were from July 1994 and 2006 (366.7 h and 358.4 h), while July 2018 had 332.1 h. Thus, the high annual mean was mainly due to a long period with constant high SDU and not just due to a few extreme monthly SDU sums. The spatial pattern of the anomalies and the long period of high SDU values might indicate long and stable high-pressure situations, which were centered over Scandinavia and Central Europe. High pressure systems in the summer half year, especially, are characterized by a low cloud fractional coverage and a high solar insolation.

## 4. Conclusions and Outlook

In this study we presented the CM SAF R Toolbox and its intention, functionality, and application. Based on the CM SAF satellite-based climate data records for SDU, it was demonstrated how the toolbox can be used to analyze large amounts of NetCDF data without any programming skills. The toolbox was used to analyze and visualize the extreme SDU in 2018 with regard to the long-term climate data record. In addition, it was demonstrated that the CM SAF SDU climate data records are well-suited for analyzing the extremes, variabilities, and changes of this important essential climate variable.

The heart of the CM SAF R Toolbox is the cmsaf R-package, which provides a wide functionality for the work with NetCDF data. A worldwide download number of over 23,500 (February 2019) and feedback from users demonstrate that the cmsaf R-package is not only useful for CM SAF data, but also for other geospatial NetCDF data.

One of the largest advantages of the CM SAF R Toolbox is that it is completely based on the scientific programming language R. R is one of the standard languages for statistical analysis of scientific data records and there is a large and steadily growing community. As with most R code, the CM SAF R Toolbox and the cmsaf R-package are free and open source. This opens the possibility for users to contribute to this software or to adapt it to their own specific needs. The CM SAF R Toolbox is designed for easy usage and its intention is to help users to get started with CM SAF NetCDF data. Although the CM SAF R Toolbox has many interactive GUI based features, users can easily access the underlying R code and adopt it for individual purposes, hence allowing advanced analyses and special solutions, which cannot be represented by the toolbox.

The CM SAF R Toolbox will be steadily developed to increase its functionality, to improve the usability and to adapt it to a wider range of NetCDF data. For instance, it is conceivable to expand its R-shiny functionalities to use the CM SAF R Toolbox for web-based processing. The implementations of more complex operators are also possible, such as the merging of satellite- and station-based data or the calculation of the solar energy potential. In addition, it is planned to implement the possibility of a selection of single countries for more user specific applications. Through this, the CM SAF R Toolbox is a useful tool for geospatial analysis and mapping for an even wider range of applications.

**Author Contributions:** Steffen Kothe developed the CM SAF R Toolbox, designed and performed the study, and analyzed the data. Uwe Pfeifroth and Jörg Trentmann contributed CM SAF sunshine duration data, provided feedback, and contributed to the CM SAF R Toolbox. Christine Träger-Chatterjee and Rainer Hollmann gave substantial feedback. All authors contributed to the concept of this study.

**Acknowledgments:** The work performed was done by using data from EUMETSAT's Satellite Application Facility on Climate Monitoring (CM SAF).

**Conflicts of Interest:** The authors declare no conflicts of interest.

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
