# Peer review of "The CM SAF R Toolbox—A Tool for the Easy Usage of Satellite-Based Climate Data in NetCDF Format"

_ijgi, doi:10.3390/ijgi8030109_

Round 1
Reviewer 1 Report
Line 132 - it's not clear what's meant by "section 2" On line 104, could you add since since when the CM SAF R Toolbox is used/developed? Lines 145 and further - could you just add a notice, how more tar files are processed (if delivery contains more tar files for one order)? Figure 5 - it would be of interest to have difference between long-term sums and 2018's sums - if possible, could you add this Fig.? Generall comment - although I understand the philosophy used for description of the workflow in the text, may be it could be of great interest of readers to have some part of it described more in deep details,, may be e.g. how the annual SDU anomaly was constructed? (lines 266 to 282). Figs. 7 and 8 - altough described in the text, it's not very clear which diagram display monthly and annual values, could it be somehow added? As for the Outlook, I think it could be quite useful to prepare spatial means not only according to coordinate selection (rectangle etc.), but also according to the countries borders - do authors intend to incorporate such feature in the near future?
Author Response
Thanks for your helpful comments. Please find our answers in the attached pdf.

Reviewer 2 Report
PDF attached - it may be more readable.
Major comments:
I found this to be an interesting and generally well-written paper.
Grammar is about 90% correct – I’ve mentioned a few of the more noticeable items below in minor comments. To be clear – this work is readable and understandable as it is, but would benefit from a native English speaker giving you a quick edit.
Table 1 – I really like this table, and think it is exceedingly useful. However, the formatting is currently detracting from this utility. Examples – there are no headers identifying function name vs description., cmsaf.add and cmsaf.div have descriptions that don’t align with them vertically, levbox_mergetime’s description is split across two pages. While the page split will hopefully be fixed in layout, vertical alignment probably wouldn’t. Something to draw the eye to the name of the functions themselves may also be useful – perhaps bolding? Similarly, a line between the data analysis description and the data manipulation names may help separate this into two sets of data in four columns, and not 1 set with 4 columns.
297-98 – given that you call out Berlin and Toulouse in your text, it may be worth noting in figure captions that their locations are shown deliberately. I had assumed it was some sort of feature of the default map used by the toolbox, and ignored them in the figures.
Figures 7 and 8 – please add significantly more to these figure captions. Captions should stand alone in describing their figures, rather than relying on the text entirely to explain them. For instance, why is there a CM SAF logo on the average seasonal cycle plot? Is it meaningful? What are the blue regions in that same plot? Etc… With 6 plots per figure, labels for sub-figures (A, B, C, etc) may be very helpful as well. This would allow you to better describe the plots in your figure caption, and let you reference Figure 7A or 8C rather than Fig 7, top left, etc.
Minor comments:
26 – “are used since” should be “have been used for”
27 – “for instance” really not needed here
35 – ditto
98 – “I.e.” can be deleted here, as it is unnecessary and a jarring way to start a sentence
110 – “fulfil” needs a 2nd L at the end
116 – no “a” before “software”, though “…to be software which allows…” (also deleting the comma) would likely convey your point more clearly
129-131 – these may look better indented
138 – “as download” doesn’t seem to fit here – I’d delete it.
138 – “the form” instead of just “form”
152 – “As with” and not “As for” (alternatively, just delete this phrase entirely and start “This script…”)
170-172 – this seems to be a repeat of the previous paragraph (section 2.2). I’d delete, move, or rephrase this.
180 – “users” should be “user’s”
184 – “form” needs a “the” before it
185 – “png file” should either be “a png file” or “png files”
205 – “parameters” may be more clear with a colon(“:”) after it.
209 – “Basis” should be “The basis”
228 – “a data amount of” is likely not really needed here
236 - “requires to: “ should likely just be “requires: “
242 - unsure where its American vs British English, or English vs other languages, but I think “12.784” should be “12,784” instead.
Figure 5 – I realize that these colors are very likely traditional for sunshine duration, but as a color-blind remote sensor reading red/yellow scaled data is an exercise in frustration. If these colors are not traditional (guessing they are given Figure 4 as well), please consider a different scale. If they are traditional, please consider this comment to just be a whiny reviewer J Figure 6 helped me significantly to follow along on your analysis of Fig 5.
259 – “north-west” should be “the north-west”
293 – should be “extraordinarily” instead
300 – “Basis” should be “The basis”
302 – “The” not needed here (actually, it is incorrect to have it)
328 – “effected” should be “affected” (English is a pain. I had to double-check myself to make sure this was correct)
358 – while “exemplarily” is an awesome word, it really doesn’t belong in science writing (or at least not here)
360 – “well-suited” may be a better term; “excellent” would need to be “excellently” if kept here (not recommended)
363 – comma vs period in numbers (line 242 comment)
368 - “As most” should be “As with most”
371 – I’d change “an easy” to just “easy”
372-3 – While your point here is true, this sentence has some grammar issues, really undercuts your arguments, and doesn’t fit within the rest of this paragraph. I’d delete it.
378 – “a web-based hosted processing” may make more sense as “web-based processing”
380 – I’d delete “By this” from this sentence
Author Response

(The authors gave the same response as above.)

Reviewer 3 Report
General comments
The paper presents the CM SAF R Toolbox – a set of R scripts to transform tar files (as delivered by CMSAF) into one netcdf file, run climate operators (from the cmsaf package), and visualize data and results (with R-shiny routines). The focus here is on the toolbox, i.e. not on the climate operators themselves.
After describing the toolbox, the authors provide an example of its application to analyse the CMSAF CDR on sunshine duration in 2018.
The intention of creating a toolbox is convincing, e.g. to support users training, to give an access to CMSAF climate data to unexperienced users, etc. The number of cmsaf R-package (30.000 downloads) reveals the large interest in the provided functionality.
In this context, the paper will be useful for the community of CMSAF data users.
I do recommend publishing the paper after addressing four main issues:
(1) The authors explain the advantage of entirely R-based toolbox. However, the known issue of R is its performance. I would suggest elaborating more on this issue. What are the possibilities to parallelize the calculations? Is the cmsaf package capable to do that? Would some operations be much faster in cdo?
(2) Since CMSAF data is stored in netcdf format, and presented toolbox also uses this format for the ‘input file’ for all calculations (cmsaf operators), it would be beneficial to add more details about the netcdf format itself.
(3) I am not convinced by the second part of the paper: the sunshine duration analysis. It is too basic to be a real ‘climatological’ analysis (which I guess was not the intention of the authors). But concurrently, it is not enough detailed to be a tutorial how to use the toolbox. I would suggest creating a step-by-step tutorial-style description of the procedure used to download, read, prepare and analyse the data and to create all the figures. This could be either provided in a supplement material to this paper, or as a vignette to the toolbox with a link given in the paper.
(4) The language must be improved. There are some lengthy sentences with peculiar structure, missing commas, missing definite and indefinite articles (the/a).
Specific comments
L17-18. “To visualize the data….” – rephrase the sentence
L23. Maybe R-project should be added to the keywords
L35. Remove ‘for instance’
L37. Or --> and
L38-39. Not clear if the resolution is of CDRs or satellite data
L44. Start new sentence with “In addition”
L47. Provide examples of other data providers in netcdf format. Not clear what ‘quai standard’ mean
L53. “NetCDF data are…” – rephrase
L56. ‘huge data amounts’ is relative. Give some examples in GB/TB
L58. Not clear what ‘aggregate’ means in this context
L66. Remove ‘especially’
L71. command-line-based
L78. above-mentioned
L87. ‘branches’ – not clear what is meant
L95. csv is also text. What about ascii – be more precise here
L98. Remove I.e.; satellite-based
L112. ‘certain system functions’ – not clear
L118. Add information about the R toolbox licence (not only that is free to download)
L138. ‘as download’ - rephrase
L139. Tar file is not a tool - rephrase
L150. There are some repetitions between 2.2 and 2.2.1. Maybe remove 2.2.1 and keep one description in 2.2. The link to cmsaf package is anyway given in the first sentence
Table1. Formatting is not clear. Mark the operators that can be used in the toolbox.
L236. ‘to choose one of the tar files’ – what about multiple files?
Author Response

(The authors gave the same response as above.)
